

# CDCP1 knockdown suppresses PDGFRβ/AKT pathway-mediated vascular smooth muscle cell proliferation by inhibiting PDGFRβ endocytosis

Xin Ji[1,2,*], Xin Wang[3,*], Qianqian Dong[3], Wanqiu Li[3], Ning Zhou[2], Xiaole Yue[3], Dandan Zhao[3] and Xiaolong Yang[1]

[1] Hebei Key Laboratory of Animal Physiology, Biochemistry and Molecular Biology, Hebei Collaborative Innovation Center for Eco-Environment, Ministry of Education Key Laboratory of Molecular and Cellular Biology, College of Life Sciences, Hebei Normal University, Shijiazhuang, Hebei Province, China

[2] Department of Clinical Laboratory, Southern University of Science and Technology Hospital, Shenzhen, Guangdong Province, China

[3] Department of Clinical Laboratory, The Second Hospital of Hebei Medical University, Shijiazhuang, Hebei Province, China

[*] These authors contributed equally to this work.

Corresponding author
Xiaolong Yang, yangxl@hebtu.edu.cn

## ABSTRACT

CUB domain-containing protein 1 (CDCP1) is a type of cell surface glycoprotein that has been identified as being capable of regulating cell anchorage, migration, proliferation, and differentiation. However, the contributions of CDCP1 in intimal hyperplasia, specifically regarding the proliferation and migration of vascular smooth muscle cells (VSMC), are unclear. In this study, we analyzed CDCP1 expression on intimal hyperplasia through a focal carotid stenosis model *in vivo*. *In vitro*, we cultured mouse VSMCs and stimulated them with 20 ng/mL platelet-derived growth factor BB (PDGF-BB) for 24 h. Western blot analysis was performed to detect the expression of CDCP1 in the cells. Next, we knocked down the expression of CDCP1 in VSMCs and assessed its effects on cell proliferation and migration using CCK8 assays, EDU[+] assay, and wound healing experiments. We then performed RNA-Seq analysis on the CDCP1-knockdown VSMCs. Based on the sequencing results, we utilized western blotting to investigate the impact of CDCP1 knockdown on the AKT signaling pathway. Additionally, we validated the interactions between Platelet-derived growth factor receptor (PDGFR)β with NEDD4, clathrin, and Rab5 using immunofluorescence and co-immunoprecipitation assays. The results discovered that CDCP1 levels were activated in the intimal hyperplasia tissues *in vivo*. CDCP1 knockdown significantly attenuated mouse VSMC proliferation and migration induced by PDGF-BB *in vitro*. Based on the Gene Ontology (GO) and Kyoto Encyclopedia of Genes and Genomes (KEGG) enrichment analysis of the differentially expressed proteins obtained from RNA-sequencing, we found that the knockdown of CDCP1 is related to the "PI3K-AKT signaling pathway", "ubiquitin-mediated proteolysis", and "endocytosis" pathways. The subsequent experiments demonstrated that CDCP1 knockdown inhibited AKT signaling pathway. CDCP1 knockdown promoted the binding of PDGFRβ and NEDD4, and PDGFRβ ubiquitin. Moreover, CDCP1 knockdown attenuated the binding of PDGFRβ to clathrin and Rab5. These data reveal that the absence of CDCP1 may

enhance the binding of PDGFR to NEDD4 and promote the ubiquitination of PDGFR, thereby regulating the AKT signaling pathway and intimal hyperplasia.

## INTRODUCTION

Intimal hyperplasia is an important pathological feature of atherosclerosis (*Ross, 1999*) and significant contributor to the unsuccessful outcomes of various vascular reconstructive interventions, including angioplasty (*Orford et al., 2000*), transplant vasculopathy (*Laks & Dipchand, 2022*), vascular stenting (*Liuzzo, Ambrose & Coppola, 2005*), and vein graft (*Newby & George, 1996*).

Platelet-derived growth factor receptor (PDGFR) is essential in the platelet-derived growth factor (PDGF) signal transduction pathway. It is a glycoprotein characterized by a single polypeptide chain that spans the membrane, categorized under the type III tyrosine kinase family and distributed in human tissues, such as smooth muscle cells, fibroblasts, endothelial cells, neuroglial cells, and chondrocytes. PDGFR, including PDGFRα and PDGFRβ, is essential for enhancing cellular proliferation, invasion, and angiogenesis. Vascular smooth muscle cell (VSMC) proliferation is induced by cytokines and growth factors, such as PDGF (*Xu, Pelisek & Jin, 2018*), a key growth factor (*Zernecke, 2014*).

The proliferation of VSMCs triggered by PDGF-BB is crucial in the progression of vascular disorders (*Wanjare, Agarwal & Gerecht, 2015*). Consequently, understanding the regulatory mechanisms underlying PDGF-BB signaling in the suppression of VSMC proliferation represents a crucial pharmacological approach for preventing intimal hyperplasia.

Under PDGF-BB stimulation, PDGFRβ undergoes a series of intracellular transport processes (*Kim et al., 2020*). First, invagination of clathrin leads to the formation of small vesicles, which are internalized by the cell (*Zhou et al., 2022*). Second, clathrin is hydrolyzed and replaced by Rab5 to transport PDGFRβ (*Hyttinen et al., 2013*). Clathrin-dependent endocytosis enhances signaling and responses (*Chen et al., 2009*).

CUB domain-containing protein-1 (CDCP1) is a membrane-associated protein characterized by the presence of three extracellular CUB domains, which serve as a substrate for the Src family of kinases (*Huang et al., 2020*). CDCP1, a new transmembrane protein, promotes angiogenesis and cell invasion; CDCP1 and IL18-R1 are elevated in individuals diagnosed with primary ventricular fibrillation (PVF) when compared to healthy subjects. Alterations in the DNA sequence influence the expression of the 3p21.31 locus (CDCP1), which is linked to myocardial infarction (*Liu et al., 2023*; *Oliveras et al., 2022*; *Shia et al., 2011*). Although CDCP1 has been characterized as a regulator in various cancers, including breast (*Roberts et al., 2001*) and ovarian (*Khan et al., 2021*), its function in intimal hyperplasia has not yet been sufficiently elucidated. In this work, we discovered CDCP1 expression was significantly higher in tissues affected by intimal hyperplasia

and aimed to elucidate the mechanism of CDCP1 in intimal hyperplasia. Knocking down CDCP1 increased the binding of PDGFRβ and NEDD4, enhancing PDGFRβ ubiquitination, resulting in reduced endocytosis and a downregulation of PDGFRβ/AKT signaling, ultimately inhibiting cell proliferation and migration.

## METHODS

### Murine focal carotid stenosis model

Animal experiments conformed to the guidelines issued by the Ethics Committee of Guangdong Medical Experimental Animal Center. Approval for this study was granted by the Ethics Committee at the Guangdong Medical Experimental Animal Center (ethical number: D202403-51). Twelve male C57BL/6 mice, 8 weeks, were randomly allocated into two equal groups and subjected to a focal carotid stenosis model, as previously described (*Kumar & Lindner, 1997*; *Mukai et al., 2006*; *Tao et al., 2013*). The mice were housed in SPF-grade animal facilities in the Comparative Medical Laboratory of the Guangdong Medical Experimental Animal Center, with three mice per cage, maintaining a controlled room temperature (24–25 °C), with a 12-h light/dark cycle. Standard feed and water were provided to ensure the health and comfort of the mice. After the administration of a 1% pentobarbital sodium *via* intraperitoneal injection to anesthetize mice, the left common carotid artery was isolated and ligated at the point of the left internal and external carotid arteries. The right common carotid artery was used as the sham group, only separated and not ligated. At 7, 14, 21, and 28 days postoperatively, overdose of 1% pentobarbital sodium by intraperitoneal injection was used for euthanasia of all mice. The bilateral common carotid arteries were obtained and slices were prepared for subsequent experiments.

### Histological and morphological analysis

For the histological analysis, the collected arteries were fixed and embedded in paraffin and subsequently sectioned into five μm sections for hematoxylin-eosin staining (Cat. no. G1076; Servicebio, Wuhan, China). First, stain with hematoxylin for 2 min, then differentiate. After differentiation, stain with eosin staining solution for 30 s, followed by another differentiation step. Finally, proceed with dehydration and clearing. The measurements of the lumen area, intimal area, and medial area were conducted utilizing ImageJ software.

### Cell culture

VSMCs derived from mice were obtained from Fenghuishengwu Inc. (cat#CL0710; Wuhan, China), cultured in DMEM (cat#C3120-0500; VivaCell, Yerevan, Armenia) supplemented with 10% FBS (cat#164210; Pricella, Houston, TX, USA) and 1% penicillin streptomycin (cat#03-031-1BCS; Boehringer Ingelheim, Ingelheim am Rhein, Germany), and kept at 37 °C in 5% $CO_2$.

### siRNA synthesis and transfection

siRNA oligonucleotides targeting the mouse CDCP1 gene (siCDCP1) were designed. Using a random sequence as a negative control. A riboFECT-TM CP Transfection

Kit (Cat. no. C10511-05; RiboBio, Guangzhou, China) was used to transfect siCDCP1 (CAGAAGCTACAGCAGCATA) into mouse VSMCs. Briefly, 50 pmol siCDCP1 with 1 X buffer and 12 µL reagent were combined for 15 min and cultured alongside VSMCs for 24 h at 37 °C with 5% $CO_2$. The interval between transfection and subsequent experiments was 24 h.

## qRT-PCR

RNA was isolated from VSMCs using the CELL RNA kit (Cat. no. 19231ES50; YEASEN, Shanghai, China). RNA quality was assessed by a microplate reader (Invitrogen, Waltham, MA, USA). Using a cDNA reverse transcription kit (Cat. no. R323; Vazyme, Nanjing, China), 1,500 ng of RNA was reverse transcribed into cDNA in accordance with the guidelines outlined in the kit manual. We then performed quantitative real-time PCR for validation. The PCR amplification protocol was as follows: 95 °C for 3 min, followed by 40 cycles of 95 °C for 15 s, 60 °C for 30 s, and 72 °C for 25 s. Finally, an extension step at 72 °C for 10 min was performed. The primer sequences used were as follows: Cdcp1: Forward: TCTCTCGGTATGGCTGGTCA, Reverse: ACGGTCTCCTTCGAGTACCA; β-actin: Forward: GTTGCTATCCAGGCTGTGCT, Reverse: GAGGGCATACCCCTCGTAGA.

## CCK8 assay

We measured cell activity with CCK8 to determine the activity in VSMCs. VSMCs were first plated into a 96-well plate, followed by VSMCs transfected with CDCP1 siRNA. VSMCs were stimulated with PDGF-BB (20 ng/mL) (Cat. no. 50611; Cell Signaling Technology, MA, USA) for another 24 h. Subsequently, CCK8 solution (Cat. no. C0037; Shanghai, China) was introduced into each well and subsequently incubated in a cell incubator for a duration of 2 h. The final results were then quantified using a microplate reader (Cat. no. 1900; Shanpu, Shanghai, China) at a wavelength of 450 nm.

## Cell proliferation assay

The proliferation of VSMCs was assessed using an EDU Cell Proliferation Kit (Cat. no. C0075S; Beyotime Biotechnology, Haimen, China). After the transfection process, EDU was added to the cells and incubated for 2 h, following the manufacturer's instructions. During each EDU assay, five random fields were analyzed under a microscope. The percentage of positive cells was calculated by dividing the number of EDU-positive cells by the total cell count in each field.

## The wound-healing assay

The wound-healing assay was employed to evaluate the migration of cells. VSMCs were placed into new culture dishes and transfected with CDCP1 siRNA. After complete cell fusion, we created a two mm scratch in the cell, washed it twice with PBS, replaced it with low-serum culture medium with fetal bovine serum (containing 1% FBS), and added PDGF-BB (20 ng/mL) to it. We then collected images under the microscope 24 h later and calculated the damage repair area of each group of cells.

## Transswell assay

A cell migration assay was conducted using an eight μm transwell plate (Cat. no. 3422; Corning, NY, USA). Cells were seeded in the upper chamber of the transwell plate at a density of $5 \times 10^3$ cells per well. The lower chamber was filled with 500 μL of culture medium containing 10% FBS. The cells were then incubated for another 24 h. Migrated cells were stained with 1% crystal violet (Cat. no. G1062; Solarbio, Beijing, China) and photographed under a microscope.

## RNA-sequencing

RNA was isolated from VSMCs using the CELL RNA kit (Cat. no. 19231ES50; YEASEN, Shanghai, China). RNA quality was assessed by a microplate reader (Invitrogen, Waltham, MA, USA). RNA sequencing were performed between the control siRNA and CDCP1 siRNA group. Differential expression analysis after CDCP1 knockdown was performed using the DESeq2 R package (1.20.0). Differentially expressed genes (DEGs) with |log 2FC| > 1 and Padj < 0.05 were recognized as genes that are significantly expressed differently. A total of 123 DEGs were identified and cluster analysis was then performed. The enrichment analysis of differential genes through Gene Ontology (GO) and Kyoto Encyclopedia of Genes and Genomes (KEGG) can elucidate the functional enrichment of DEGs and delineate distinctions at the level of gene functionality. In this study, we employed the clusterProfiler package in R to conduct both GO functional enrichment and KEGG pathway enrichment analyses. The statistical power associated with this experimental design, as determined by RNASeqPower, is 0.96. A *P*-value of less than 0.05 was deemed indicative of significant enrichment in the GO or KEGG pathways.

## Western blot

Harvested cells were lysed using RIPA Lysis Buffer (Cat. no. R0010; Solarbio, Beijing, China). Protein extracts (30 μg of total protein/sample) were electrophoresed using 10% SDS-PAGE (Cat. no. MA0387; Meilunbio, Dalian, China) and moved onto PVDF membranes (Cat. no. 03010040001; Sigma, Burlington, MA, USA). The membranes that were blotted underwent a blocking procedure using 5% non-fat milk for a duration of 1 h. Following this, they were incubated at 4 °C overnight with primary antibodies targeting CDCP1 (Cat. no. 12754-1-AP; Proteintech, Wuhan, China), Rab5 (Cat. no. 46449; Cell Signaling Technology, Danvers, MA, USA), PDGFRβ (Cat. no. ET1605-20; Huaan, Hangzhou, China), AKT (Cat. no. ET1609-51; Huaan, Hangzhou, China), p-PDGFRβ (Cat. no. 3166; Cell Signaling Technology, Danvers, MA, USA) and P-AKT (Cat. no. AF0016; Affinity Biosciences, Cincinnati, OH, USA). The membranes underwent an additional incubation with secondary antibodies conjugated to horseradish peroxidase for a duration of 1 h. Subsequently, the signals were visualized employing an electrochemical luminescence (ECL) reagent. Images were acquired using a chemiluminescence imaging system JP-K600 (Jiapeng, Guangdong, China).

## Co-immunoprecipitation

For co-immunoprecipitation (Co-IP), cell lysis was performed using protein lysis buffer (150 mM NaCl, 50 mM Tris-HCl, pH 8.0, 1.0% NP-40, 0.5% sodium deoxycholate, and

0.1% SDS) supplemented with a comprehensive proteinase inhibitor (Cat. no. P0100; Solarbio, Beijing, China) according to the manufacturer's instructions. Antibodies against PDGFRβ (Cat. no. ET1605-20; Huaan, Hangzhou, China) were added to the lysates with protein A/G beads (Cat. no. sc-2003; Santa Cruz, CA, USA) and kept overnight at 4 °C. After centrifugation, the precipitate was collected and cleaned three times. After resuspending the mixture with 2X loading buffer, we heated at 95 °C for 5 min. The compounds were re-centrifuged and the supernatant was used for western blotting.

For the ubiquitination experiment, cells were lysed, and PDGFRβ antibody (Cat. no. ET1605-20; Huaan, Hangzhou, China) along with protein A/G beads (Cat. no. sc-2003; Santa Cruz, CA, USA) were mixed with the lysate for overnight incubation at 4 °C. The following day, the bound proteins were analyzed using western blotting with a ubiquitination antibody (Cat. no. sc-8017; Santa Cruz, CA, USA).

## Immunofluorescence

Paraffin-embedded five μm sectioned common carotid arteries were incubated with CDCP1 antibody (Cat. no. EM1701-84; Huaan, Hangzhou, China). Mouse VSMCs were cultured on confocal dishes for 24 h. For immunofluorescence staining, the cells were subjected to fixation and permeabilization utilizing a solution of 0.1% Triton X-100 (Cat. no. G1076; Solarbio, Beijing, China) at room temperature for 30 min, and exposed to PDGFRβ (Cat. no. ab69506; Abcam, Cambridge, UK), NEDD4 (Cat. no. 2740; Cell Signaling Technology, Danvers, MA, USA), clathrin (Cat. no. 4796; Cell Signaling Technology, Danvers, MA, USA), Rab5 antibodies (Cat. no. 3547; Cell Signaling Technology, Danvers, MA, USA) at 4 °C overnight. The cells were treated with Alexa Fluor@555 conjugated Goat Anti-mouse IgG (Catalog number 1030-32; SouthernBiotech, Birmingham, AL, USA) and Alexa Fluor@488 conjugated Goat Anti-rabbit IgG (Catalog number 4030-30; SouthernBiotech, Birmingham, AL, USA) at room temperature for a duration of 1 h, protected from light. Finally, nuclei were additionally stained with DAPI (Cat. no. 0100-20; SouthernBiotech, AL, USA). Pictures were obtained utilizing a fluorescence microscope (BZ-X810, KEYENCE) at 600x magnification.

## Statistical analysis

All statistical analyses were conducted employing Prism 9.0 software. To assess the significance between the two groups, an unpaired Student's $t$-test was utilized, while one-way analysis of variance, succeeded by Tukey's test, was implemented to determine differences across multiple groups. The results are expressed as mean $\pm$ standard deviation. A $P$-value of less than 0.05 was deemed statistically significant.

## RESULTS

### Increased CDCP1 expression in intimal hyperplasia *in vivo*

To identify whether CDCP1 participated in intimal hyperplasia, we ligated the bifurcation of the internal and external carotid arteries in mice. Sections of the common carotid artery obtained at 7, 14, 21, and 28 days were taken separately after ligation, and hematoxylin and eosin (HE) staining was used to examine intimal hyperplasia in mice. Intimal hyperplasia

gradually increased with prolonged ligation times in mice, and after 21 days of ligation, the blood vessels were completely blocked (Fig. 1A). We measured the vascular lumen area, intimal area, and medial area and calculated the intimal/medial ratio at each time point separately. The lumen area gradually decreased (Fig. 1B), while the intimal and medial area gradually increased (Figs. 1C and 1D), and the intimal/medial thickness ratio also gradually increased (Fig. 1E). Additionally, immunofluorescence staining of the ligated carotid artery showed that the expression levels of CDCP1 were markedly elevated in comparison to the sham group, and positively correlated with the degree of intimal hyperplasia (Figs. 1F and 1G). Morever, the result of the western blot was consistent with those of the IF analysis. With the extension of the ligation days, the expression of CDCP1 increased and reached its highest level in the tissue after 21 days of ligation (Figs. 1H and 1I).

## CDCP1 knockdown inhibited VSMCs proliferation and migration triggered by PDGF-BB

Since arterial blood vessels are primarily composed of smooth muscle cells, endothelial cells, and fibroblasts, macrophages are involved when inflammation occurs. We selected four different types of cells (VSMCs, ECs, fibroblasts, macrophages) to detect CDCP1 expression, and found VSMCs had the highest expression of CDCP1 (Figs. 2A and 2B). Next, we stimulated VSMCs using PDGF-BB (20 ng/mL) for 24 h, and discovered CDCP1 level was significantly upregulated (Figs. 2C and 2D). These results proved CDCP1 was involved in cell proliferation in VSMCs.

To further understand the role of CDCP1 within cellular contexts, CDCP1 siRNA was implemented to transfect VSMCs, qRT-PCR was leveraged to assess the knockdown efficiency of CDCP1, and the results showed that CDCP1 expression was significantly knocked down (Fig. 2E). Abnormal proliferation and migration of VSMCs are major contributors to intimal hyperplasia. Therefore, we continue to investigate the effects of CDCP1 knockdown on the proliferation and migration of VSMCs. CCK-8 and EDU assays were utilized to evaluate cell viability after CDCP1 knockdown, showing that CDCP1 knockdown significantly inhibited cell viability (Fig. 2F) and EDU+ cells (Figs. 2G and 2H) under PDGF-BB stimulation. The Transwell and wound healing assays were performed to confirm CDCP1's effect on cell migration, and we found CDCP1 knockdown inhibited cell migration (Figs. 2I and 2J, Fig. S2). Similarly, PCNA expression was decreased under PDGF-BB stimulation after CDCP1 knockdown (Figs. 2K and 2L). These results indicated that knocking down CDCP1 inhibited cell proliferation and migration in VSMCs.

## RNA-Seq analysis of CDCP1 knockdown in mouse VSMCs

We confirmed CDCP1 was involved in cell proliferation and migration; however, its regulatory mechanism remained unclear. To this end, CDCP1 was knocked down in mouse VSMCs for RNA sequencing. A total of 123 differential genes were identified based on RNA sequencing, in which 25 genes exhibited upregulation and 98 genes exhibited downregulation (Fig. 3A). These DEGs were then studied for GO enrichment and KEGG pathway enrichment. GO enrichment analysis revealed CDCP1's function on "negative regulation of cellular response" and "protein phosphatase binding" (Fig. 3B). KEGG

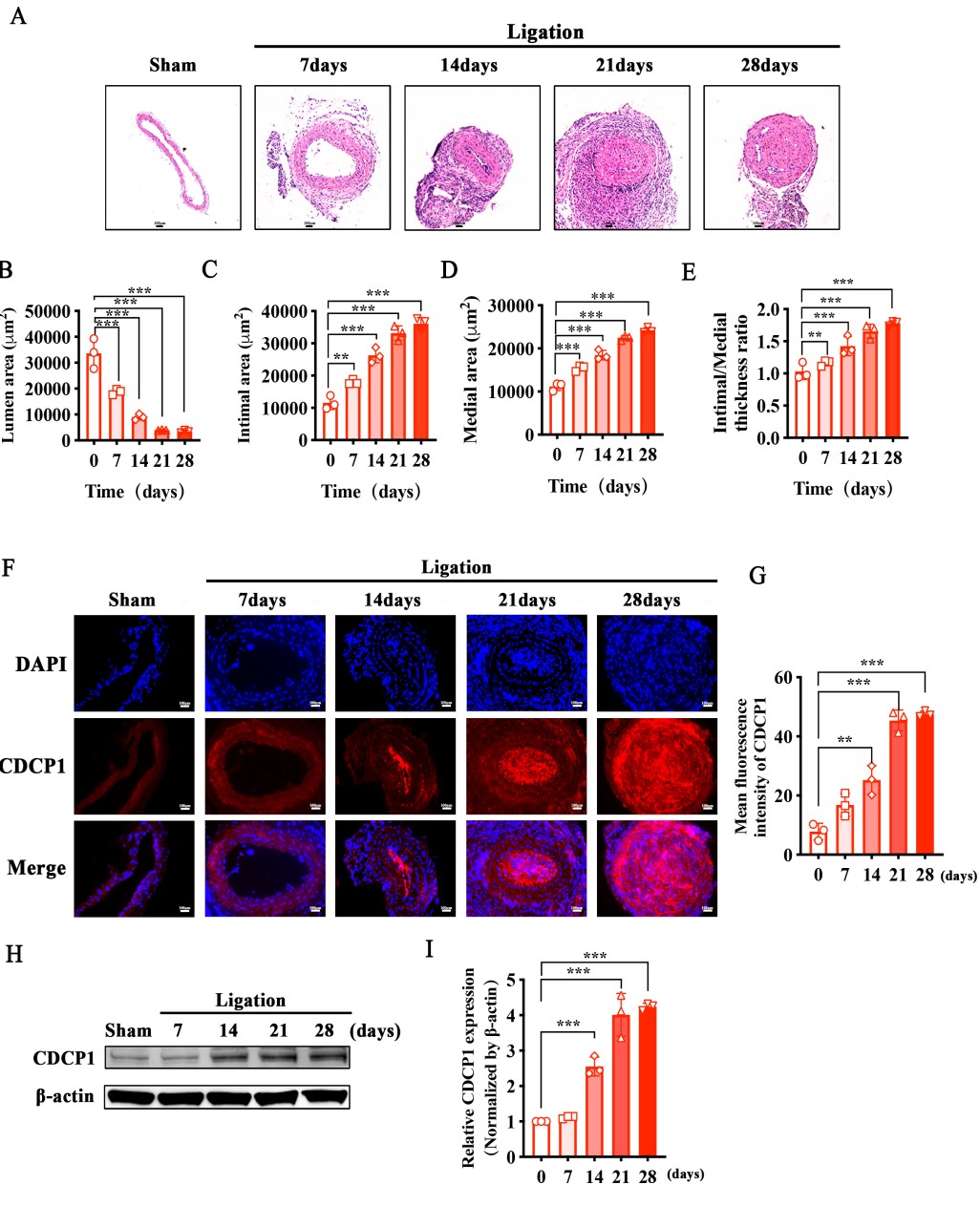

**Figure 1  Increased CDCP1 expression in intimal hyperplasia *in vivo*.** (A) On the 7th and 14th day after injury, the intima of the carotid artery was apparently thickened, as revealed by Hematoxylin and Eosin staining. (B–E) Statistical analysis of lumen area (B), intimal area (C), medial area (D) and intimal/-medial thickness ratio (E). (F and G) Immunofluorescence staining of CDCP1 in intimal hyperplasia. * indicates $P < 0.05$, ** indicates $P < 0.01$, and *** indicates $P < 0.001$.

enrichment analysis indicated CDCP1's function on "PI3K-AKT signaling pathway", "ubiquitin mediated proteolysis", "endocytosis", and "MAPK signaling pathway" (Fig. 3C).

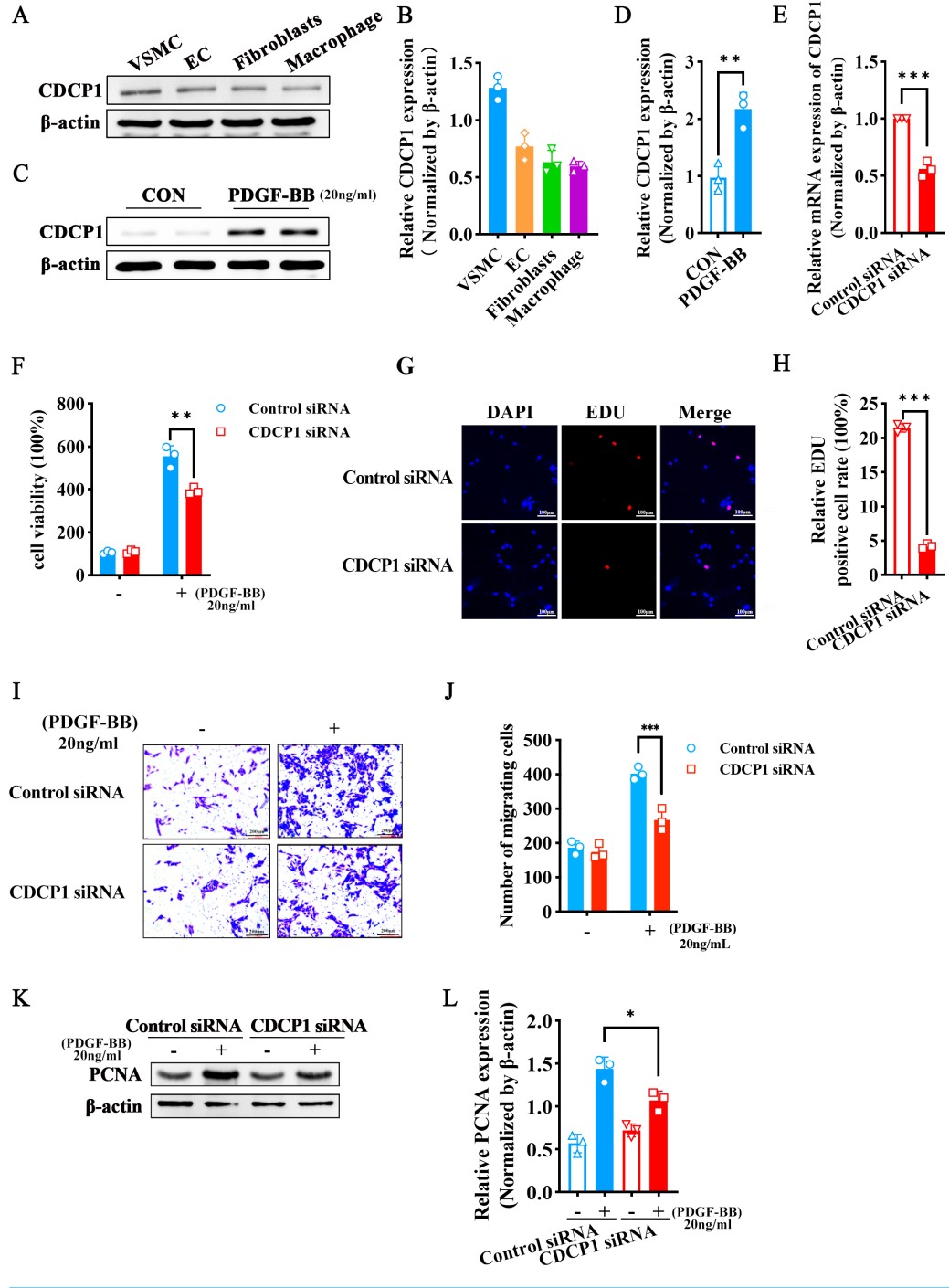

**Figure 2  CDCP1 knockdown inhibited cell proliferation and migration.** (A) CDCP1 expression in VSMCs, ECs, fibroblasts and macrophage. (B) Quantitative analysis of CDCP1 in different cell types. (C) Western blot was used to detect CDCP1 expression under (continued on next page...)

**Figure 2 (...continued)**
PDGF-BB stimulation 24 h in VSMCs. (D) Quantitative analysis of CDCP1 with PDGF-BB stimulation. (E) Cell viability of VSMCs stimulated with PDGF-BB was inhibited following CDCP1 knockdown in a Cell Counting kit-8 assay. (F) EDU$^+$ staining for cell proliferation detection after CDCP1 knockdown. (G) Statistics on the number of EDU positive cells in each group. (H and I) In a wound-healing assay of VSMCs after CDCP1 knockdown. (J) Western blot was used to detect PCNA expression under PDGF-BB stimulation for 24 h after CDCP1 knockdown. (K) Quantitative analysis of PCNA. Data were expressed as mean ± SD. *$P < 0.05$ indicated a statistically significant difference. * indicates $P < 0.05$, ** indicates $P < 0.01$, and *** indicates $P < 0.001$.

## CDCP1 knockdown inhibited PDGFRβ/AKT signaling in VSMCs induced by PDGF-BB

The AKT signaling pathway plays a critical role in regulating the proliferation of smooth muscle cells. Activation of the AKT pathway can lead to increased cell proliferation. To assess the role of CDCP1 in regulating AKT signaling, mouse VSMCs transfected with CDCP1 siRNA were serum-starved for 12 h and afterward exposed to PDGF-BB for 5, 15, and 30 min, after which PDGFRβ phosphorylation and downstream AKT signaling were determined by western blot (Figs. 3D–3G). The results suggested that CDCP1 expression was significantly decreased with CDCP1 siRNA treatment (Fig. 3E). Phosphorylated PDGFRβ (Fig. 3F) and AKT (Fig. 3G) levels also decreased with PDGF-BB (20 ng/mL) stimulation for 30 min after CDCP1 knockdown. Moreover, we also examined the effect of CDCP1 knockdown on the ERK pathway. With the prolonged stimulation of PDGF-BB, the phosphorylation level of ERK gradually increased. However, knockdown of CDCP1 significantly inhibited the activation of ERK phosphorylation (Fig. S1). These results indicate that knocking down CDCP1 inhibits the PDGFRβ/AKT signaling pathway and the ERK signaling pathway.

## CDCP1 knockdown increased NEDD4-mediated PDGFRβ ubiquitination

RNA-seq analysis revealed CDCP1's function in ubiquitin mediated proteolysis (Fig. 3C). To assess whether CDCP1 knockdown had an impact on the ubiquitination of PDGFRβ, we conducted immunoprecipitation experiments to detect changes in the PDGFRβ ubiquitination level after CDCP1 knockdown in VSMCs. The findings indicated that the reduction of CDCP1 expression markedly elevated the ubiquitination rate of PDGFRβ under the condition of PDGF-BB stimulation (Fig. 4A). NEDD4 is a ubiquitinase that serves a key function in governing cellular processes including protein degradation, membrane trafficking, and cell signaling (*Yang et al., 2020*). Therefore, we examined whether CDCP1 knockdown affected the expression of NEDD4 and the binding of PDGFRβ to NEDD4. Co-IP results demonstrated CDCP1 knockdown increased the association of PDGFRβ with NEDD4 under PDGF-BB stimulation (Figs. 4B and 4C). However, the relationship of PDGFRβ and NEDD4 was not affected in the absence of PDGF-BB stimulation (Figs. 4B and 4C). Therefore, we conducted immunofluorescence (IF) assay to further explore the binding of PDGFRβ and NEDD4 under PDGF-BB stimulation. The results also showed CDCP1 knockdown enhanced the co-localization of PDGFRβ and NEDD4 (Figs. 4D and 4E). This result further corroborates the impact of CDCP1 on protein ubiquitination as

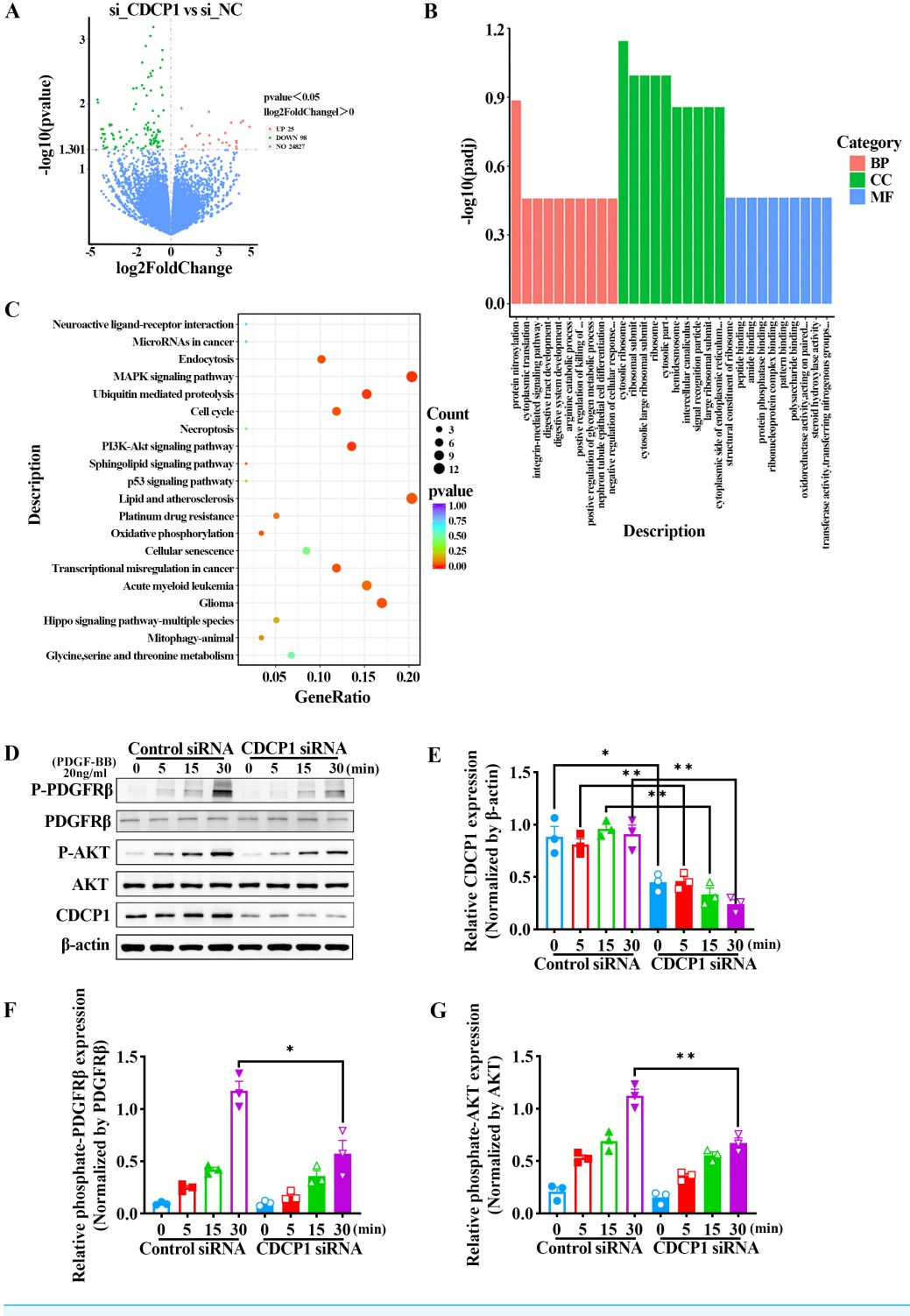

**Figure 3** **RNA-seq analysis indicated decreased PDGFRβ/AKT signaling in VSMCs stimulated with PDGF-BB after CDCP1 knockdown.** (A) Differential genes analysis after CDCP1 knockdown based on RNA-seq. (B) GO enrichment analysis of differential genes. (continued on next page...)

**Figure 3 (…continued)**
(C) KEGG enrichment analysis of differential genes. (D) Western blot was used to detect the phosphory-lation level of PDGFRβ and AKT under PDGF-BB stimulation at indicated times (0, 5, 15, 30 min). (E–I) Quantitative analysis of CDCP1 (E), PDGFRβ (F), phosphate-PDGFRβ (G), AKT (H), phosphate-AKT (I). Data were expressed as mean ± SD. *$P < 0.05$ indicated a statistically significant difference. * indicates $P < 0.05$, ** indicates $P < 0.01$, and *** indicates $P < 0.001$.

indicated by the RNA-Seq analysis and explains the experimental findings that CDCP1 knockdown inhibits the activation of the PDGFRβ/AKT signaling pathway.

**CDCP1 knockdown inhibited clathrin-mediated PDGFRβ endocytosis**

PDGFRβ is primarily expressed on the cell membrane. Upon stimulation with PDGF-BB, PDGFRβ is internalized into the cell, where it undergoes subsequent processes such as signaling pathway activation or degradation. Under PDGF-BB stimulation, PDGFRβ was internalized into cells mediated by clathrin (*Kim et al., 2020*). Clathrin-dependent endocytosis enhances signaling and responses (*Chen et al., 2009*). To investigate how CDCP1 exerts its functions in VSMCs stimulated by PDGF-BB, Co-IP, and IF assays were conducted to verify the interaction between CDCP1 with clathrin. The Co-IP results indicated CDCP1 knockdown attenuated the co-localization of PDGFRβ and clathrin under PDGF-BB stimulation (Figs. 5A and 5B). In order to provide additional confirmation regarding the impact of CDCP1 knockdown on the interaction between PDGFRβ and clathrin under PDGF-BB stimulation, we used IF to locate PDGFRβ (red) and clathrin (green) in VSMCs stimulated by PDGF-BB and found that CDCP1 knockdown decreased the interaction of clathrin and PDGFRβ (Figs. 5D and 5E).

After PDGFRβ coated by clathrin internalized in the cell, clathrin is hydrolyzed and replaced by Rab5 to transport PDGFRβ and mediate its intracellular functions (*Hyttinen et al., 2013*). The result of Co-IP assay showed that the relationship of PDGFRβ and Rab5 was inhibited under PDGF-BB stimulation (Figs. 5A and 5C). Furthermore, IF assay also demonstrated that PDGFRβ (green) interacted with Rab5 (red), and CDCP1 knockdown reduced the relationship between Rab5 and PDGFRβ under PDGF-BB treatment (Figs. 5F and 5G). These results suggest CDCP1 knockdown leads to reduced PDGFRβ endocytosis, which may inhibit the PDGFRβ signaling pathway.

## DISCUSSION

The activation of the PDGFRβ signaling pathway is crucial for intimal hyperplasia (*Fredriksson, Li & Eriksson, 2004*). Its expression and activity are influenced by multiple factors, such as synthesis, endocytosis, recycling, and degradation. This study revealed CDCP1 as a new regulator of PDGFRβ activity. In the absence of CDCP1, ubiquitination of PDGFRβ was enhanced, and PDGFRβ endocytosis was inhibited, resulting in the downregulation of the PDGFRβ/AKT signaling pathway and cellular dysfunction. However, the relationship between PDGFRβ ubiquitination and PDGFRβ endocytosis has not been explored in depth in this study. Therefore, we will continue to investigate the regulatory relationship between PDGFRβ ubiquitination and endocytosis in future research.

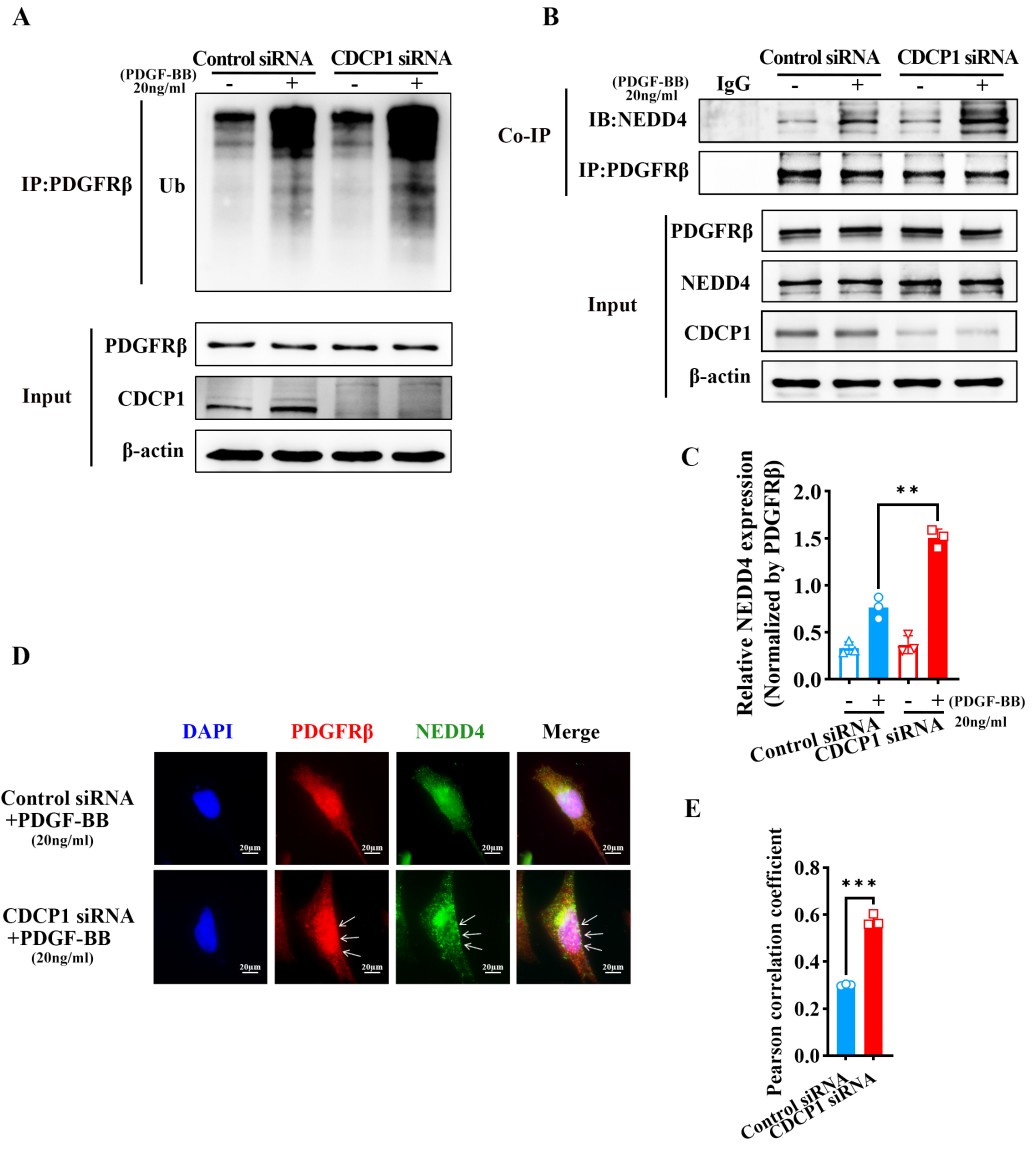

**Figure 4** **CDCP1 knockdown increased the binding of NEDD4 and PDGFRβ.** (A) Co-immunoprecipitation assay was used to demonstrate the interaction between PDGFβ and NEDD4 after CDCP1 knockdown. (B) Quantitative analysis of NEDD4. (C) Cells were treated with 20 ng/mL PDGF-BB for 30 min, and stained. Proteins were detected with PDGFRβ (red) and NEDD4 (green) antibodies. (D) Quantitative analysis of confocal/NEDD4 (green). Data were expressed as mean ± SD. *$P < 0.05$ indicated a statistically significant difference. * indicates $P < 0.05$, ** indicates $P < 0.01$, and *** indicates $P < 0.001$.

This study revealed an increased expression of CDCP1 in intimal hyperplasia *in vivo* and examined the possible mechanisms of CDCP1 in mouse VSMCs. Further *in vivo* studies are needed to interfere with CDCP1 expression in order to clarify its impact on the regulation of intimal hyperplasia. The construction of intimal hyperplasia model has certain limitations. The focal carotid stenosis is a simplified murine intimal hyperplasia model, which is similar to the complete ligation model in that it is likely to create the disturbed hemodynamic forces

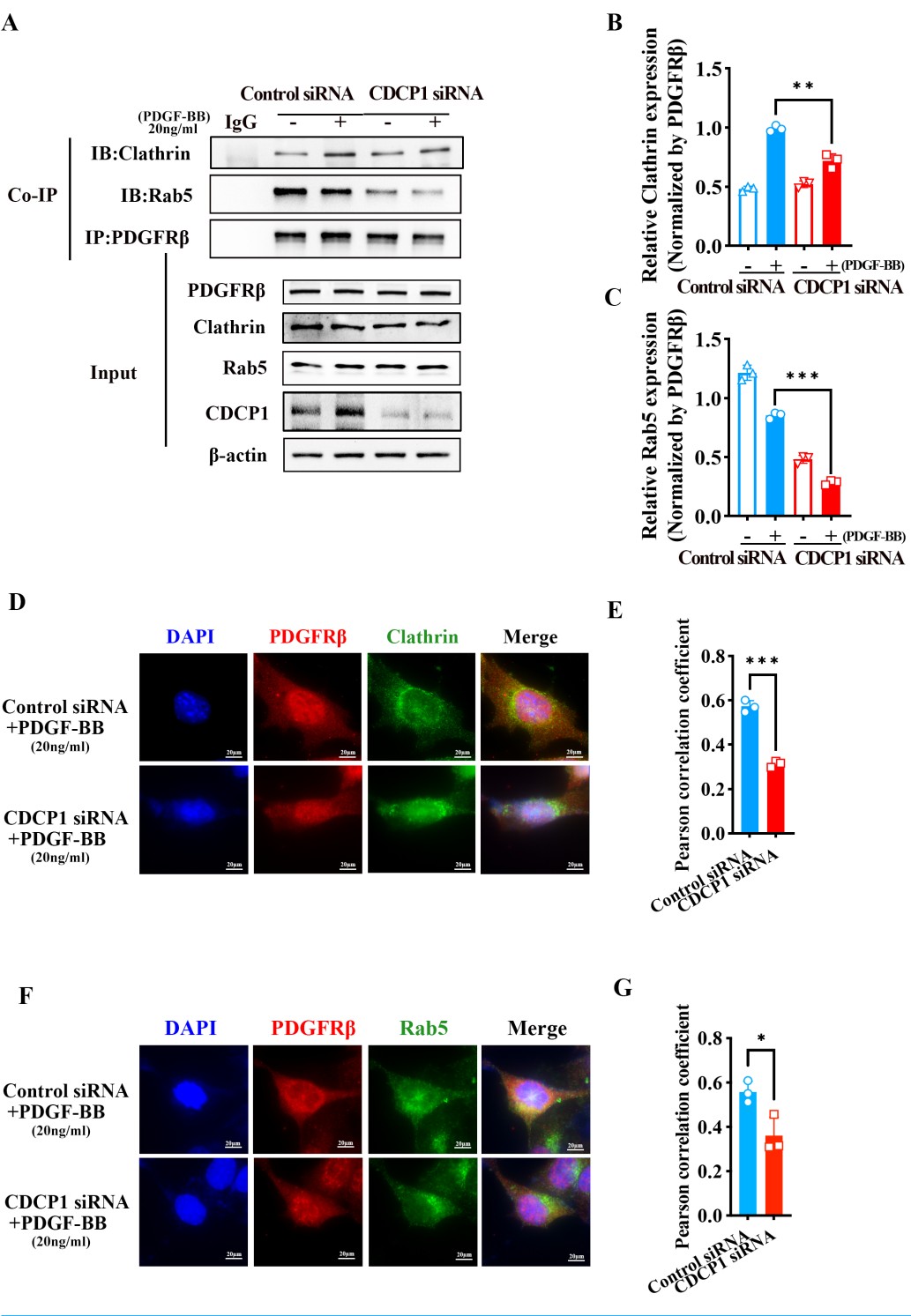

**Figure 5   CDCP1 knockdown inhibited Clathrin mediated PDGFRβ endocytosis.** (A and C) Cells were treated with 20 ng/mL PDGF-BB for 30 min, and stained. The relationship between PDGFRβ (red) and Clathrin (green), PDGFRβ (red) and Rab5 (green) was measured by IF. (continued on next page...)

that subsequently drive the intimal hyperplasia response, rather than directly mimicking an anatomical situation (*Tao et al., 2013*). However, based on RNA-seq analysis, we discovered CDCP1's influence on abnormal lipid metabolism and atherosclerosis. Therefore, we will continue to use ApoE$^{-/-}$ mice to build a hyperlipidemia-induced arterial atherosclerosis model to verify the role of CDCP1 in lipid metabolism.

After confirming that CDCP1 expression is upregulated during intimal hyperplasia, we further investigated its effects on cell proliferation and migration by knocking down CDCP1 in cultured smooth muscle cells *in vitro*. To assess cell migration, we employed both wound healing and Transwell assays to validate the impact of CDCP1 on cell migration. In the wound healing assay, we reduced the serum concentration in the cell culture medium to 1% FBS to minimize the influence of serum on the migration results. However, even under low serum conditions, cells continued to proliferate. Therefore, we also utilized the Transwell assay to further verify the influence of CDCP1 on cell migration. Our results indicated that knocking down CDCP1 inhibited cell migration.

The Rab family contains more than 70 Rab proteins belonging to 44 subfamilies, of which small GTPases, the most representative family, participate in the complex regulation of membrane transport (*Tao et al., 2013*). Five Rabs (Rab1, Rab5, Rab6, Rab7, and Rab11) are expressed in many eukaryotes and play essential roles (*Ao, Zou & Wu, 2014*; *Ohbayashi & Fukuda, 2012*). Rab5 is located in early endosomes and plays a role in mediating the fusion of endocytic vesicles to generate early endosomes (*Langemeyer, Frohlich & Ungermann, 2018*). Our results showed that CDCP1 contributed to the development of intimal hyperplasia. Furthermore, *in vitro* knockdown of CDCP1 decreased cell proliferation and affected the intracellular transport of PDGFRβ through the clathrin-Rab5 pathway.

We found that CDCP1 could regulate PDGFRβ endocytosis mediated by clathrin. CDCP1 knockdown decreased the binding of clathrin to PDGFRβ. Additionally, Rab7 is a common endocytotic Rab protein that coordinates the transport between late endosomes and lysosomes and is involved in autophagy maturation (*Kofler et al., 2018*). Moreover, Rab4 positive endosome participates in PDGFR recycling to membrane (*Liu et al., 2023*), However, it is not yet known whether CDCP1 affects the Rab4-mediated process of receptor recycling to the cell membrane. Therefore, we need to continue to explore the mechanism of CDCP1's influence on PDGFRβ endocytosis and intimal hyperplasia from multiple perspectives in order to provide a new treatment strategy for atherosclerosis in the future.

## CONCLUSION

This study showed that CDCP1 expression is elevated in intimal hyperplasia tissues *in vivo*. CDCP1 knockdown enhanced the PDGFRβ ubiquitination and inhibited PDGFRβ

endocytosis induced by PDGF-BB, leading to reduced PDGFRβ/AKT signaling pathway, which ultimately inhibited VSMC proliferation and migration *in vitro*.

## ACKNOWLEDGEMENTS

The authors are grateful to all the experts who gave them guidance in the experiments.

### Funding

This research was funded by the Shenzhen Nanshan District Health Major Project (NSZD2024069). The funders had no role in study design, data collection and analysis, decision to publish, or preparation of the manuscript.

### Grant Disclosures

The following grant information was disclosed by the authors:
The Shenzhen Nanshan District Health Major Project: NSZD2024069.

### Competing Interests

The authors declare there are no competing interests.

### Author Contributions

- Xin Ji conceived and designed the experiments, performed the experiments, analyzed the data, prepared figures and/or tables, authored or reviewed drafts of the article, and approved the final draft.
- Xin Wang performed the experiments, analyzed the data, prepared figures and/or tables, and approved the final draft.
- Qianqian Dong performed the experiments, analyzed the data, authored or reviewed drafts of the article, and approved the final draft.
- Wanqiu Li performed the experiments, prepared figures and/or tables, and approved the final draft.
- Ning Zhou performed the experiments, authored or reviewed drafts of the article, and approved the final draft.
- Xiaole Yue analyzed the data, prepared figures and/or tables, and approved the final draft.
- Dandan Zhao analyzed the data, prepared figures and/or tables, and approved the final draft.
- Xiaolong Yang conceived and designed the experiments, authored or reviewed drafts of the article, and approved the final draft.

### Animal Ethics

The following information was supplied relating to ethical approvals (*i.e.*, approving body and any reference numbers):

Approval for this study was granted by the Ethics Committee at the Guangdong Medical Experimental Animal Center (ethical number: D202403-51).

## DNA Deposition

The following information was supplied regarding the deposition of DNA sequences:

The CDCP1 sequences are available at GenBank: NM_022842.

## Data Availability

The raw measurements are available in the Supplementary File.

## Supplemental Information

Supplemental information for this article can be found online at http://dx.doi.org/10.7717/peerj.19114#supplemental-information.

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
