# Peer review of "CDCP1 knockdown suppresses PDGFRβ/AKT pathway-mediated vascular smooth muscle cell proliferation by inhibiting PDGFRβ endocytosis"

_PeerJ, doi:10.7717/peerj.19114_

## Round 0.1 · original submission · Major Revisions

Although reviewers found the topic is interesting, they also raised serious concerns. The authors need to address all their concerns carefully including professional language editing so that the manuscript will be re-evaluated for potential publication.

Reviewer 1 ·

Basic reporting

The quality of English used is poor throughout the article. There are tons of spelling mistakes and many sections have incoherent sentence composition.

Experimental design

Lack of proper controls and incorrect data interpretation.

Validity of the findings

Reviewer comments can be found in the annotated PDF.

Annotated reviews are not available for download in order to protect the identity of reviewers who chose to remain anonymous.

Reviewer 2 ·

Basic reporting

In this study, the authors aimed to identify a new target, CUB domain-containing protein 1 (CDCP1), in promoting SMCs phenotypic transitions and pathogenesis of intimal hyperplasia (IH). The author also demonstrated that this process was achieved by activating PDGFRβ/AKT pathway. This manuscript has good novelty and data quality, yet I recommend addressing several issues to improve the manuscript's quality, as delineated in the subsequent comments.

Experimental design

Figure1: the authors detected CDCP1 level in murine carotid stenosis model (AKA complete ligation model). However, the duration of this model usually lasts 28 days after ligation surgery. It is still plausible that at 28 days, CDCP1 has no changes between sham and ligation groups. But I am wondering how CDCP1 level changes at different time courses. And I didn’t see authors clarify what time spot is used for CDCP1 IF staining.
Moreover, since the ligation model is the only animal model established in this manuscript, n=3 is a relatively small n number. I would recommend 5 or more mice should be conducted for CDCP1 staining.

Figure 2: I noticed the authors showed siCDCP1 KD efficiency in the following figures. However, the evidence that siCDCP1 significantly reduced CDCP1 protein level should be presented in fig.2.
In this figure, the authors show the result of cell viability and wound heal assay. However, the methods of these 2 assays were not described. In contrast, the authors described a cell invasion assay, which has no results showing through the manuscript.

In figure 2 and 3: please indicate the concentration of PDGF-BB used in figure 2 and 3.

Validity of the findings

Fig 4: the binding of PDGFRβ and NEDD4 is not direct evidence for PDGFRβ degradation and ubiquitination. Please prove PDGFRβ is truly degraded or ubiquitinated.

Additional comments

There are some missed literatures which the authors should cite. In line 62/63, Liu D Circ Res. 2023 (PMID: 37800334) should be mentioned and cited. In line 231, literatures should be cited to support the claim.

Please revise the manuscript again and improve the language, correct typos and formats. Such as: in line 249 “nternalized”, aberrant spaces in line 215, 220.

Reviewer 3 ·

Basic reporting

The manuscript "CDCP1 knockdown suppresses PDGFRβ/AKT pathway mediated Vascular Smooth Muscle Cells proliferation by inhibiting PDGFRβ endocytosis " evaluated the CDCP1 expression in hyperplasia, and its function of regulating the PDGFRβ-NEDD4 interaction for endocytosis, thus provide new insights into the prevention and treatment of vascular diseases. Most of the results are clear and credible, but there are still some issues for improvement.
Major:
1. The manuscript is well, concisely and coherently organized and presented and the style, but language and grammar need to be accurate and appropriated. Especially in the abstract, the language description of the methods and the results is too poor.
2. The author has been emphasizing the function of CDCP1 phosphorylation, but there is no detection of CDCP1 phosphorylation throughout the article.
3. The statistical method should be written in the figure legends, and the pictures and statistical charts that have been labeled separately need to be written separately.
4. The statistical results are inconsistent with the trend of representative pictures, like Figure 2I.
5. Figure 3 is too blurred to make the result clear.
6. Incorrect interpretation of fluorescence colocalization coefficient and IB in Co-IP. Such as Figure 4B, 4D, 5B, 5D, 5F, 5G.

Minor:
1. The western blot results of increased CDCP1 expression in intimal hyperplasia in vivo should be produced.
2. The number of references can be appropriately increased.
3. The figures need to be trimmed. Put the bars together, not interspersed. Then the width of the bar chart should be consistent.
4. Some formatting errors need to be noted, such as improper skipping on line 220.

Experimental design

The manuscript is well, concisely and coherently organized and presented and the style.

Validity of the findings

Most of the results are clear and credible, but there are still some issues for improvement.

Additional comments

The upper and lower cases of the letters in the title may require careful consideration.

---

## Round 0.2 · Major Revisions

Please carefully and adequately address the reviewer's comments.

Reviewer 2 ·

Basic reporting

The manuscript looks great now. All my concerns have been addressed and I have no more question regarding the manuscript.

Experimental design

All my concerns have been addressed and I have no more question regarding the experimental designs.

Validity of the findings

All my concerns have been addressed and I have no more question regarding the data

Reviewer 3 ·

Basic reporting

After the revision, the authors have responded meticulously to the comments provided by the reviewers and are thanked for their diligent efforts in enhancing the scientific merit and publication readiness of the article. However, I still harbor three primary concerns.

The initial sentence of the abstract, which is intended to encapsulate the background and research significance of the entire paper, mentions CDCP1 regulating cellular functions through phosphorylation. In reality, the paper elaborates on CDCP1's role in regulating the phosphorylation of PDGFR. Consequently, there exists a significant discrepancy in the expression used, which undermines the clarity and accuracy of the abstract.

Secondly, given the profound impact of PDGF-BB on the proliferation of smooth muscle cells, the wound healing assay fails to elucidate its effect on the migration function of these cells. The healing area observed in the wound-healing assay could solely be attributed to cellular proliferation, thereby limiting the scope of the conclusion to describe the influence on cell migration.

Thirdly, regarding the endocytosis of PDGFR, the fluorescence images depicted in Figures 6A and 6C, under identical treatment conditions, exhibit a striking difference in the localization distribution of PDGFR. This inconsistency undermines the reliability of the results presented. Furthermore, the experimental outcomes of Co-IP indicate that PDGF-BB does not stimulate the binding between Rab5 and PDGFR, casting doubt on the occurrence of its endocytosis.

Additionally, there are minor issues such as the inconsistent bolding of qRT-PCR in the methodology section. Moreover, in the discussion, it is advisable to refrain from exaggerating CDCP1's ability to promote intimal hyperplasia, as the corresponding treatment has not been conducted or substantiated within the scope of this study.

Experimental design

no comment

Validity of the findings

no comment

---

## Round 0.3 · Minor Revisions

The Section Editor has provided the following detailed feedback, which should be addressed to improve the manuscript:

1. Incorporation of Reviewer Comments: Several significant answers to the reviewers have not been integrated into the manuscript. The methodology section, in particular, needs to be clarified. For example, the manuscript does not mention the following important detail: “In our scratch assay, we opted for a low-serum culture medium with fetal bovine serum (containing 1% FBS), which significantly inhibited cell proliferation, thereby reflecting cell migration capacity to some extent.” This should be added, and the potential limitation of this approach should be addressed in the discussion.

2. Methods and Materials Section: The Methods and Materials section needs to be completed for both transparency and reproducibility. Specific details are missing, such as relevant reference codes (e.g., the CELL RNA kit from Yeasen) and the proteinase inhibitor used in the co-immunoprecipitation (co-IP) experiment. Additionally, clarification is needed on whether the PDGFRβ and PDGFR antibodies are the same. The description of the qRT-PCR procedure is also insufficient—key details are missing, particularly regarding primers and qRT-PCR conditions.

3. Control Inconsistencies in IF and co-IP: There are inconsistencies in the controls for the immunofluorescence (IF) and co-IP data, which could undermine the interpretation of the treatment results. The co-localization assays, in particular, need further clarification. This is an important question that many readers will likely have, and the authors should address it comprehensively in the manuscript.

4. Experimental Design and Integration of CDCP1: The experimental design appears fragmented, lacking clear integration of the protein CDCP1, its mechanism, and its functional outcomes. This fragmentation leads to confusion regarding the different levels of expression of CDCP1. Multiple assays are conducted, but the results are not properly interpreted in relation to one another. This lack of cohesion diminishes the manuscript's overall relevance and coherence, as it fails to clearly demonstrate how CDCP1 interacts with its mechanisms to drive a specific function.

In summary, further revisions are required before the manuscript can be considered for publication. The authors should carefully address these comments, ensuring that the methodology is clearly presented, the results are well integrated, and the experimental design is cohesive.

Reviewer 3 ·

Basic reporting

I am appreciated to the authors for their revisions and replies. The manuscript is good enough to be published.

Experimental design

I have no more question regarding the experimental designs.

Validity of the findings

I have no more question regarding the data.

---

## Round 0.4 · accepted · Accept

The concerns have been addressed.